# Plasma Oscillatory Pressure Sintering of Mo-9Si-8B Alloy with ZrB_2_ Addition

**DOI:** 10.3390/ma15072387

**Published:** 2022-03-24

**Authors:** Xiangyu Ding, Zhenping Guo, Xiangrong Li, Zhuoyue Li, Xin Li

**Affiliations:** 1Fundamentals Department, Air Force Engineering University, Xi’an 710051, China; pluto_70@126.com (Z.G.); lixiangrong0925@126.com (X.L.); lz_980512@163.com (Z.L.); lx202006t10@163.com (X.L.); 2School of Aircraft Engineering, Nanchang Hangkong University, Nanchang 330034, China

**Keywords:** oscillatory sintering, Mo-9Si-8B alloy, ZrB_2_ addition

## Abstract

Oscillatory pressure sintering is a novel crystal refinement technology. The doping of different concentrations of ZrB_2_ under oscillatory sintering technology (9 Hz) is discussed here, focusing on its macroscopic mechanics and oxidation resistance. In particular, doping 2.5 wt% ZrB_2_ can effectively increase the hardness of the alloy, slightly increase the fracture toughness of the alloy and have an outstanding effect on the oxidation resistance of the alloy at 1300 °C, achieving the effect of reducing mass loss by 80.3%.

## 1. Introduction

The excellent high-temperature properties of Mo-Si-B alloys have attracted much attention in the past three decades, and they have a great prospect in replacing nickel-based alloys and the static rings of high-pressure turbine guide [1,2,3]. Molybdenum is a typical refractory metal with a high melting point of about 2870 K, but its oxidation resistance is poor, so it is difficult to use it as a high-temperature structural material alone [4]. However, adding a silicon element to form silicide phase MoSi_2_ can form a protective silicon glass oxide film at high temperature, which gives the material system excellent high-temperature oxidation resistance [5]. In addition, adding 1% boron to form a borosilicate layer can also greatly improve the oxidation resistance of molybdenum at 800 K-1500 K [6]. In particular, the Mo-Si-B alloy contains the microstructure of α-Mo phase (Mo solid solution), Mo_3_Si and Mo_5_SiB_2_ (T_2_) compound phases, and its melting point can reach 2270 K [7]. In its microstructure, α-Mo phase is regarded as a ductile structure, and its volume fraction and morphology play an important role in the room temperature fracture toughness, but it is not conducive to improving the oxidation resistance and high-temperature creep resistance. Mo_3_Si phase and Mo_5_SiB_2_ phases, as brittle structures, play the role of a strengthening phase, which can improve the oxidation resistance and high-temperature creep resistance of materials, but is not conducive to enhancing the fatigue properties and fracture toughness [8,9]. The Si:B ratio of 9:8 can keep the balance between the α-Mo phase and the intermetallic phase to achieve the best mechanical properties and oxidation resistance [10]. Yan Jianhui [11] proposed the method of strengthening and toughening the bimodal grain size Mo-12Si-8.5B alloy by adding nano-ZrO_2_ (Y_2_O_3_) particles. However, the development of Mo-Si-B alloy still has the following problems: (1) Mo-Si-B alloys prepared by a powder metallurgy process have high porosity and low density, and usually there are many unclosed pores, which leads to the loss of mechanical toughness. (2) During the preparation process, oxygen impurities segregate to the boundary to reduce the bonding force of the grain boundary, causing brittleness in the alloy. In order to solve the above problems, many studies have been carried out. However, most of the research has not changed the problem of high porosity in traditional powder metallurgy [12]. The oscillatory sintering method recently proposed by Xie Zhipeng [13,14] can be used to solve the defects of traditional powder metallurgy for ceramic materials. Great progress has been made in the preparation of ceramic materials with high density, fine grains, high strength and high reliability. Dynamic pressure can make the powder particles slip and rearrange, and can also make the agglomerates fully depolymerize, compress the pores and increase the density. Therefore, this method also has great research value for solving the porosity problem of Mo-Si-B alloy.

What needs to be added in particular is that, in our previous study [15], the oscillatory frequency of 9 Hz resulted in a gain in alloy grain refinement and an improvement of mechanical properties. Furthermore, due to its excellent high-temperature stability, ZrB_2_ has been widely used in SiC ceramic materials and zircaloy-4 alloys to improve its high-temperature oxidation resistance by increasing the viscosity and stability of the SiO_2_ layer [16,17]. Specially, in Guojun Zhang’s research, although Zr or ZrB_2_ were added, the density of the alloy could only be maintained between 94% and 95%, and its high porosity was closely related to the limitations of its hot-pressing process. The porosity had a major impact on the mechanical properties of the alloy. In our previous research [15], we achieved up to 97.78% relative density of the Mo-Si-B alloy through oscillatory sintering technology. In this work, an oscillatory sintering process was used with a frequency of 9 Hz for obtaining high-density alloy. Moreover, the addition of ZrB_2_ in the Mo-Si-B alloy was designed for better property. Therefore, a detailed investigation was performed to explore the mechanical mechanism and characterize the oxidation behavior of a Mo-Si-B alloy via oscillatory pressure sintering with the addition of ZrB_2_. Our goal was to obtain a better performing Mo-Si-B alloy with ZrB_2_ added using oscillatory sintering technology.

## 2. Materials and Experimental Procedure

The samples with a nominal composition of Mo-9Si-8B (at.%) were prepared from Mo, Si and B of 99.9%, 99.5% and 99.5% purities, respectively (the following is abbreviated as MSB). Moreover, their particle sizes were ≤5 μm, 2~3 μm and ≤5 μm. The ZrB_2_ powders used had a particle size of less than 50 nm with a purity of 99.9%. The mixed powders were placed into a planetary ball with a speed of 300 rmp and a powder-to-ball weight ratio of 1:10 for 6 h to obtain a powder mixture. Then, referring to the experimental procedure of B. Li [7,18], the powders were placed into graphite mould and compacted in a vacuum environment at 1200 °C for 1 h so as to eliminate the internal stress of the powder and promote the interfacial reaction. The last step was to adopt the method of oscillatory sintering: a constant axial pressure of 40 MPa, a temperature of 1600 °C, an oscillatory pressure of 5 MPa and a sintering time of 6h were applied, as shown in Figure 1. In particular, the oscillatory sintering frequency was 9 Hz. In addition, the mass fraction of ZrB_2_ in the MSB alloy was designed as four components: 0 wt%, 0.5 wt%, 1.5 wt% and 2.5 wt%, respectively, as shown in Table 1.

The samples were determined by the Archimedes method, and the porosity was determined by mercury porosimetry. The samples were cut into Φ10 mm × 5 mm sizes. The hardness and fracture toughness were measured by the Vickers indentation method. The Vickers hardness test uses a force of 20 kg with a loading duration of 15 s. The phase was determined by X-ray diffraction (XRD), the corrosion solution was Keller’s etchant (an aqueous solution of 10 vol.% potassium ferricyanide and 10 vol.% sodium hydroxide) and the polishing method was vibration polishing. In order to determine the oxidation resistance, cyclic oxidation experiments at 1300 °C for 15 h were performed.

## 3. Results and Discussion

### 3.1. XRD and Microstructural Features of Alloy via Plasma Oscillation Sintering

Figure 2 shows the X-ray diffraction patterns of alloys with different Zr contents. It can be seen from the Figure that all microstructures contained three phases: an α-Mo phase, a Mo_3_Si phase and a T_2_ phase. It is a remarkable fact that an m-ZrO_2_ diffraction peak was observed in the Mo-9Si-8B-2.5 wt% alloy. However, no ZrB_2_ phase was found. This indicates that ZrB_2_ was decomposed to Zr, and the m-ZrO_2_ phase was formed. The microstructures of four alloys are given in Figure 3. The polishing of the grain showed light and dark color differences under the action of corrosion. According to the effect of corrosive liquid and our previous research [19], it was pointed out that the α-Mo phase was light phase, the Mo_3_Si phase was grey phase and the T_2_ phase was dark phase. In addition, as depicted in Figure 3, it can be clearly seen that as the content of ZrB_2_ increased, the degree of refinement of the structure was higher.

In order to accurately assess the grain size, we used the Archimedes section method as our measurement.
$$n=N^{2}/{L_{t}}^{2}$$
n—Number of grains per unit area, *N*—Number of grains counted, *L_t_*—length of test lead.

As shown in Figure 4, the size of the α-Mo phase dropped from 2.6 μm to 1.3 μm, which was close to the general size reduction. This fully verified that the addition of ZrB_2_ helped heterogeneous nucleation during the sintering process, and that this was more conducive to the formation of grain boundaries, reducing grain growth during the sintering process. As for the 0.6 μm ultrafine microstructure reported in a publication from Guojun Zhang [20], we conjecture that this was largely related to the particle size of the initial powder. Under the unified process, the addition of ZrB_2_ seems to contribute to crystal fineness.

### 3.2. Porosity and Density Evaluation

As shown in Figure 5, the density of the alloy was identified. The theoretical density calculation formula refers to the method used in previous studies [21]:(1)ρalloy=1(ωα−Moρα−Mo)+ωMo3SiρMo3Si+ωMo5SiB2ρMo5SiB2+ωZrB2ρZrB2

$\omega_{\alpha -Mo}$, $\omega_{Mo3Si}$, $\omega_{Mo5SiB2}$ and $\omega_{ZrB2}$ are the proportions of each phase. $\rho_{\alpha -Mo}$, $\rho_{Mo3Si}$, $\rho_{Mo5SiB2}$ and $\rho_{ZrB2}$ are the densities of each phase.

The calculated results are given in Figure 6. It can be concluded from the results that the density of the alloy fluctuates with increases in doping concentration, but the relative density increases from 97.78% to 98.75%. This was due to the decrease in the theoretical density value. However, the density results confirmed that the densities of four alloys were maintained at relatively high levels. At the same time, when adding Zr at the cost of α-Mo, the density of the alloy decreases. The relatively high density, on the one hand, is the effect of oscillatory sintering on breaking agglomerates, and on the other hand, thanks to the effect of 24 h high-energy ball milling, the powder has high surface energy.

A porosity analysis was also conducted. It can be seen from Figure 6 that the change of porosity was consistent with the change of density predicted in Figure 5. The apparent porosities of the alloys from in this article were all lower than 1%, indicating that the 9 Hz oscillatory sintering frequency was effective in suppressing the porosity.

### 3.3. Fracture Toughness and Hardness Evaluation

The hardness test was performed under the condition of 20 Kg load for 15 s. It can be clearly seen from Figure 7 that as the amount of doping increased, its contribution to the hardness of the alloy gradually increased. In particular, the hardness of the alloy doped with 2.5 wt% ZrB_2_ was increased by up to 14.9%. This was for the following two reasons. First, the addition of ZrB_2_ had a refinement effect on the alloy structure, which increased the strength of the alloy; its corner was a “soft” α-Mo. Secondly, the phase grain size was reduced by nearly half, which can be considered according to the Hall–Petch relationship. On the other hand, it was also due to the increase in the density of the alloy.

The fracture toughness of the alloy was measured by increasing the experimental loading force (load 30 Kg), which is consistent with our previous article [15]. The fracture toughness values are shown in the Table 2. This was consistent with our previous research conditions. It was found that ZrB_2_ had a certain effect on the fracture toughness of the alloy, which was increased by 7.6%. We suspected that doping ZrB_2_ helped to improve the intergranular bonding force, purify the grain boundaries and help retard crack propagation. The fine grain effect made the crack propagation tortuous, consumed more energy and improved the fracture toughness.

### 3.4. Oxidation Behavior of Mo-Si-B Alloy at 1300 ℃

Cyclic oxidation was performed at 1300 °C, which was similar to the process of repeatedly starting and stopping the engine. At the same time, in order to explore the effect of the application of shock pressure and the ZrB_2_ trace element on the oxidation resistance of the alloy, the alloys’ sintering frequency at 0 Hz, 3 Hz, 6 Hz and 9 Hz were added here for comparison. The experimental results are plotted in Figure 8. Firstly, it can be concluded that under different oscillation frequencies, the oxidation resistance of alloys at high oscillation frequencies (9 Hz) was better than that of low-oscillation-frequency sintered alloys. This was because in our previous research high-oscillation-frequency alloys showed a fine structure [15]. Moreover, the defects were greatly suppressed. It was generally believed that the fine structure can form a borosilicate protective layer faster, and that fewer defects can reduce the intrusion of oxygen. Secondly, under the same 9 Hz oscillatory frequency sintering, the alloys with a small amount of ZrB_2_ showed a higher level of oxidation resistance, and as the amount of doping increased, the oxidation resistance was stronger. The mass loss of the Mo-Si-B-2.5ZrB_2_ alloy after 15 h oxidation was −28 mg/cm^2^, while the mass loss of the Mo-Si-B-9Hz alloy was −141.9 mg/cm^2^ (mass loss reduced by 80.3%) and the mass loss of the Mo-Si-B-0Hz alloy without oscillatory pressure was −271 mg/cm^2^ (mass loss reduced by 89.7%), which shows that the microstructure was obviously refined (especially the refinement of the α-Mo phase) so that it can improve the oxidation resistance of the Mo-Si-B alloy. The Mo-Si-B alloy doped with 2.5 wt% ZrB_2_ not only further refined the structure, but also improved the protective ability of the oxide film.

By fitting the dependence of Δ*W*/*A* on time *t*, the oxidation kinetics of different alloys are analyzed [22,23]:(2)$$\frac{\Delta W}{A}={K_{t}}^{n}$$

In the equation, where Δ*W*/*A* is the weight loss per unit area, *t* is the oxidation, *K* is power law rate constant, and *n* is the power law exponent.

The results for the n value of different alloys showed that: (i) the n values of Mo-Si-B-0Hz, Mo-Si-B-3Hz, Mo-Si-B-6Hz and Mo-Si-B-9Hz were 0.96, 0.82, 0.72 and 0.69, respectively, suggesting that the oxidation behaviors do not follow linear, parabolic or cubic laws. However, it can be concluded that the oxidation rate tended to slow down. (ii) The n values of Mo-Si-B-0.5ZrB_2_, Mo-Si-B-1.5ZrB_2_ and Mo-Si-B-0.5ZrB_2_ were 0.59, 0.37 and 0.34. The oxidation kinetics of doped alloys at 9 Hz oscillatory frequency were closer to parabola (0.59) and cubic (0.34). This suggests that doped ZrB_2_ contributes greatly to the oxidation resistance rate of the alloy. We suspected that the oxide generated at 1300 °C could effectively act as a path to hinder oxygen diffusion, reduce the oxidation rate and improve the protection.

In order to better explore the relationship between the cyclic oxidation time and the change of the oxide layer, we further observed the oxide cross-section and analyzed its oxide composition. The distribution of the outermost borosilicate layer, the intermediate layer and the matrix can be clearly found in all alloys. The oxidation cross-section of the alloy at 1300 °C is shown here. The thickness of the oxide layer of the alloy was detected at 60 min and 900 min, respectively. Oxidation cross-sections showed different shapes at different time scales. As time progressed, the depth of oxygen intrusion became deeper and deeper. The MSB_0ZrB_2_ alloy in particular showed a more complicated protective layer after 1h, and the thickness reached 52.3 μm. Moreover, between the borosilicate layer and the intermediate layer appeared a very small separation, as shown in Figure 9a. When the oxidation time reached 900 min, the separation reached the degree of exfoliation shown in Figure 9e. The middle molybdenum oxide layer in Figure 9k shows a loose porous structure which does not have the ability to protect the substrate and which accelerates the intrusion of oxygen, thereby reducing the adhesion between the borosilicate layer and the substrate layer. The oxide layer of the MSB_0ZrB_2_ alloy experienced relatively severe oxidation in the initial hour, showing greater mass loss, which can be verified in Figure 9. The thickness of the oxide layer added with ZrB_2_ was less than that of the MSB_0ZrB_2_ alloy, and the protective layer generated was relatively uniform, covering the surface of the substrate completely. There was also no peeling of the oxide layer caused by uneven cyclic oxidative stress within 900 min.

The thickness of the borosilicate layer was calculated at an average of 10 points for each sample, and the statistics are shown in Table 3. It can be clearly seen that the thickness of the oxide layer of the MSB_0ZrB_2_ alloy reached 85.5 μm after 900 min, which was almost double the thickness of the oxide layer of the MSB_2.5ZrB_2_ alloy doped with 2.5 wt% ZrB_2_. This shows that oxygen invades deeper positions of the substrate, which also causes the violent performance of oxidative mass loss. This may be due to the addition of B, which improves the flow effect of the protective layer and speeds up the coverage rate of the borosilicate layer. It is worth noting that through EDS analysis, ZrO_2_ and ZrSiO_4_ were also found in the cross-section, as oxides in the borosilicate layer, which is the same as the result found in [24,25]. In short, the preparation of 2.5 wt%-doped alloy under the oscillating sintering process has excellent performance and can effectively block the intrusion of oxygen; the formed Zr oxide can also hinder the entry of oxygen. The addition of B increases the flow rate of the oxide layer and promotes protective layer flow, so that the thickness of the oxide layer is greatly reduced and has a better protective effect.

## 4. Conclusions

In this paper, the macroscopic mechanical properties and oxidation resistance performance of ZrB_2_-doped Mo-Si-B alloys were discussed, and all alloys were fabricated via the plasma oscillatory sintering technology (oscillatory frequency: 9 Hz). Doped alloys showed better performance under the preparation of grain refinement technology, which is worthy of further study. The following conclusions were drawn:(1)On the basis of fine-grain technology (9 Hz oscillatory frequency), the doping of ZrB_2_ can refine its structure and enable it to reach half the size of the undoped alloy. At the same time, the apparent porosities of the doped alloys were all controlled below 0.7%, reflecting the characteristics of high density.(2)Doping ZrB_2_ can improve the hardness and fracture toughness of the alloy, and it will continue to increase with increases in the doping amount. This was the effect not only of fine-grain strengthening, but also of ZrB_2_ purifying the grain boundary and improving the intercrystalline bonding force.(3)The non-doped oscillating sintered alloy (0, 3, 6, 9 Hz) and the alloy doped with ZrB_2_ (0.5 wt%, 1.5 wt%, 2.5 wt%: 9 Hz) were subjected to cyclic oxidation experiments at 1300 °C. The experimental results show that as the oscillatory frequency increased, the oxidation resistance was improved, and the mass loss of the alloy doped with ZrB_2_ was much better than that of the undoped alloy (mass loss was reduced by at least 80.3%), indicating that the use of oscillatory sintering technology (9 Hz) and the alloy doped with ZrB_2_ can effectively block oxygen invasion, greatly improving antioxidant performance.

## Figures and Tables

**Figure 1 materials-15-02387-f001:**
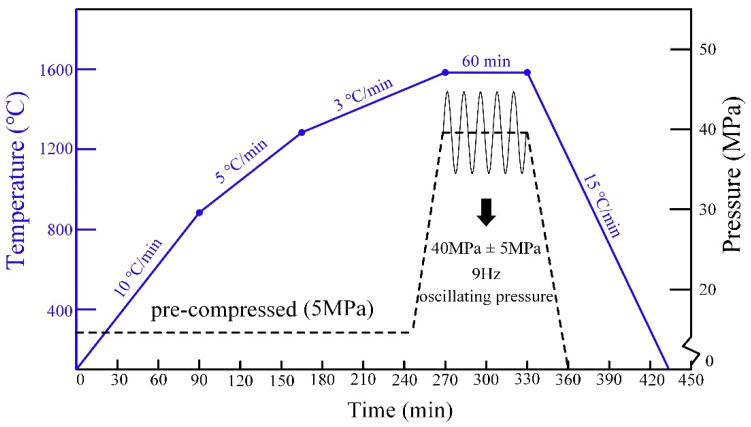
Schematic diagram of sintering mechanism in this study [19]. “Reproduced with permission from Guo Zhenping, Journal of Alloys and Compounds; published by Elsevier, [2021]”.

**Figure 2 materials-15-02387-f002:**
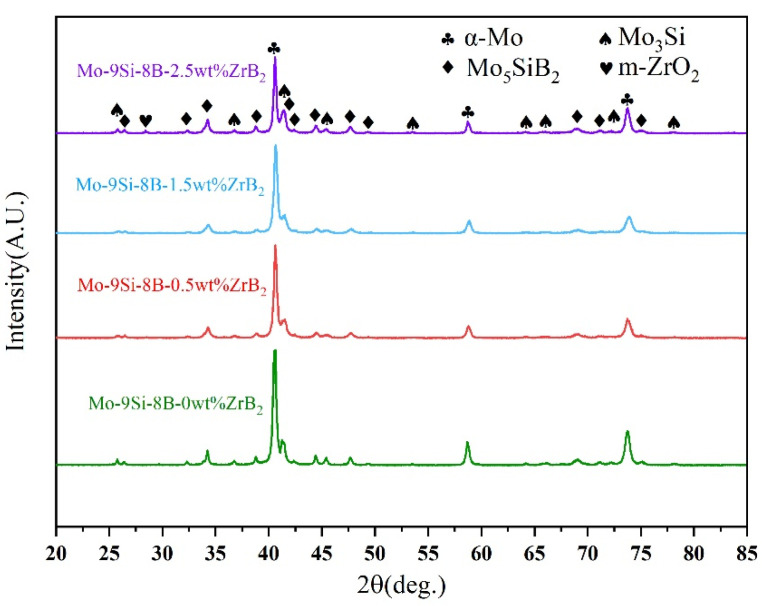
XRD diffraction patterns for alloys with different Zr contents.

**Figure 3 materials-15-02387-f003:**
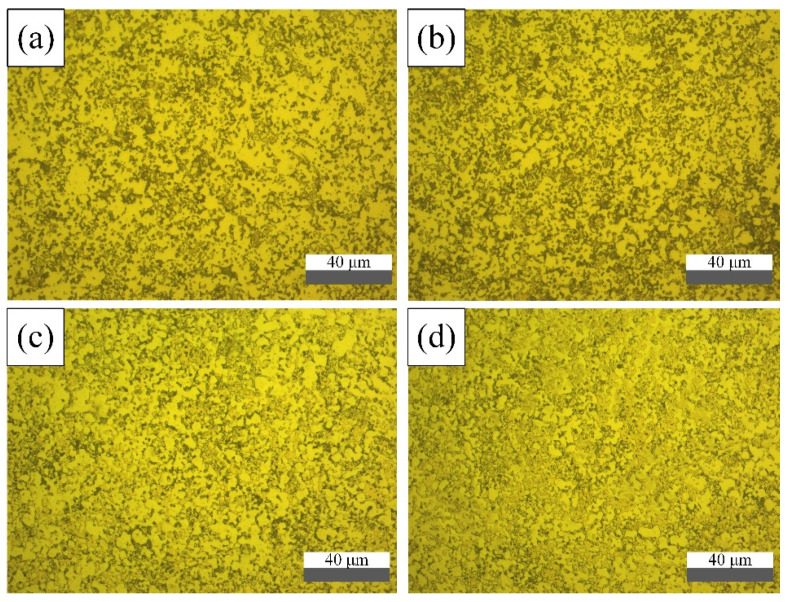
Optical micrograph of alloys with different Zr contents: (**a**) MSB_0 alloy, (**b**) MSB_0.5 alloy, (**c**) MSB_1.5 alloy and (**d**) MSB_2.5 alloy.

**Figure 4 materials-15-02387-f004:**
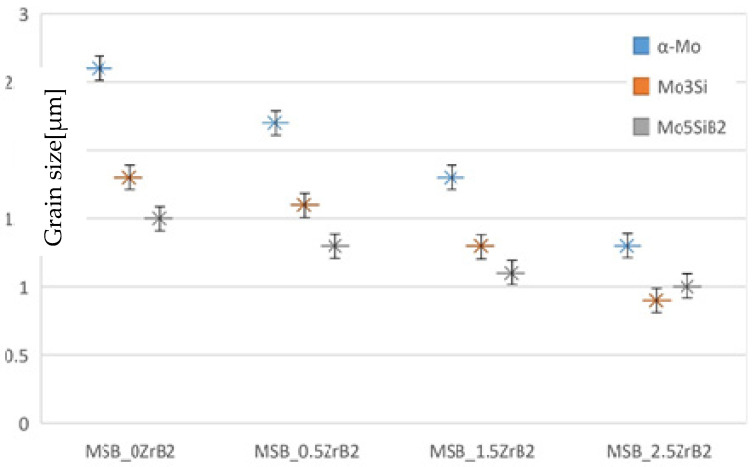
Grain size diagram of alloys with different Zr contents.

**Figure 5 materials-15-02387-f005:**
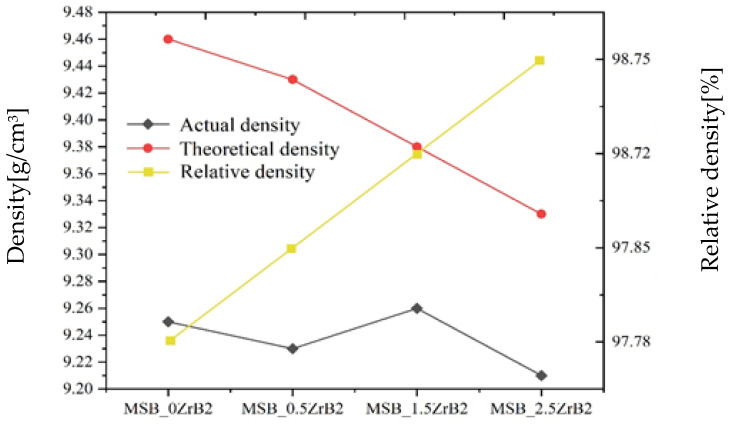
Actual density, theoretical density and relative density of alloys.

**Figure 6 materials-15-02387-f006:**
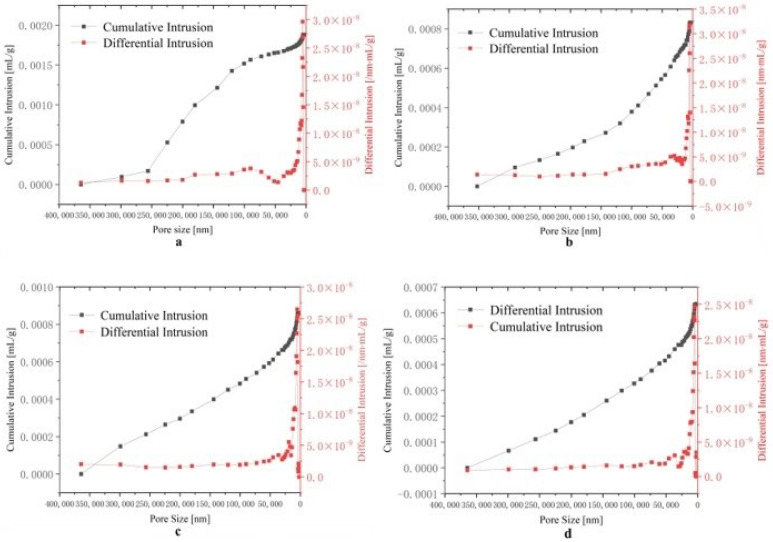
Mercury injection curves of four alloys: (**a**) MSB_0 alloy, (**b**) MSB_0.5 alloy, (**c**) MSB_1.5 alloy and (**d**) MSB_2.5 alloy.

**Figure 7 materials-15-02387-f007:**
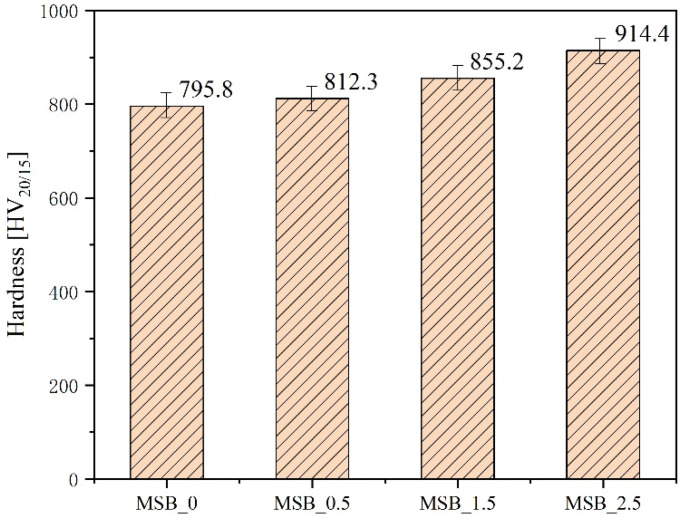
Hardness of alloys with different Zr contents.

**Figure 8 materials-15-02387-f008:**
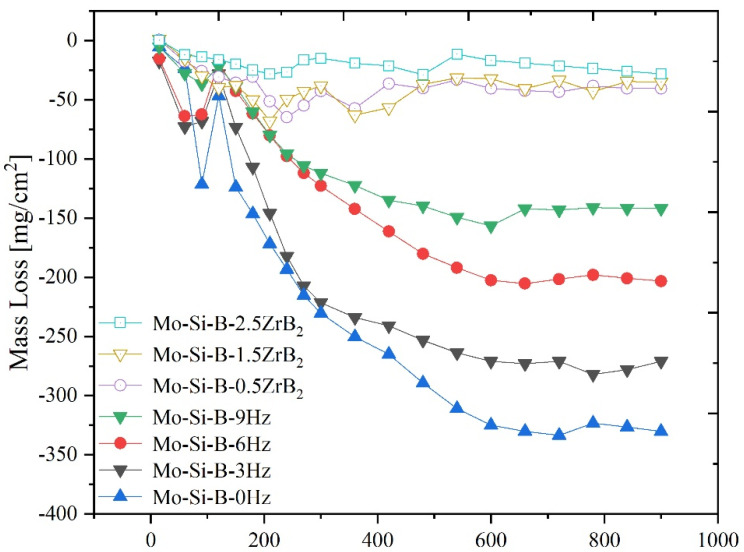
Oxidation mass loss of alloys at different frequencies and different doping levels.

**Figure 9 materials-15-02387-f009:**
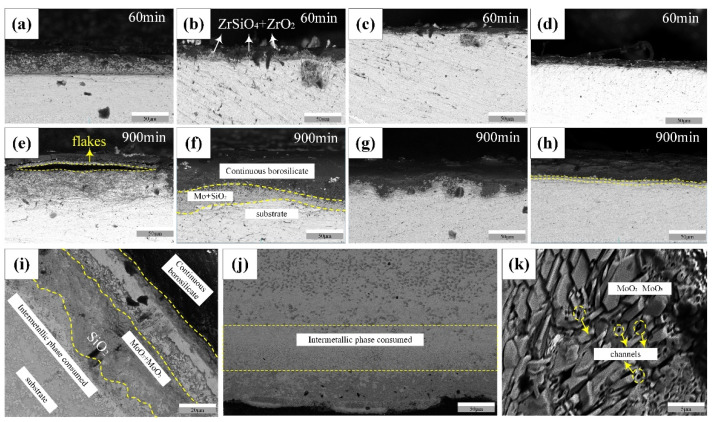
Oxidation profile at different time scales: (**a**,**e**) MSB_0ZrB_2_, (**b**,**f**) MSB_0.5ZrB_2_, (**c**,**g**) MSB_1.5ZrB_2_, (**d**,**h**) MSB_2.5ZrB_2_, (**i**) MSB_0.5ZrB_2_ alloy oxide layer delamination; (**j**) The consumption of intermetallic phase of the MSB_0ZrB_2_ alloy and (**k**) molybdenum oxide product of the MSB_0ZrB_2_ alloy.

**Table 1 materials-15-02387-t001:** Investigated alloys with different additions.

Mo-9Si-8B (at.%)	MSB_0ZrB_2_	MSB_0.5ZrB_2_	MSB_1.5ZrB_2_	MSB_2.5ZrB_2_
Content (wt%)	0	0.5	1.5	2.5
Pressure (MPa)	40 ± 5	40 ± 5	40 ± 5	40 ± 5
Frequency (Hz)	9	9	9	9

**Table 2 materials-15-02387-t002:** Indentation, crack length and K_c_ for selected alloys.

Sample Number	MSB_0.5ZrB_2_	MSB_1.5ZrB_2_	MSB_2.5ZrB_2_
Size of indentation (μm)	255.45 ± 4.3	250.35 ± 2.6	243.37 ± 4.2
Size of crack length (μm)	303.85 ± 5.8	390.79 ± 2.4	287.01 ± 5.3
K_c_ (MPa √m)	11.31	11.77	12.17

**Table 3 materials-15-02387-t003:** Calculated borosilicate layer thickness for selected alloys.

Average Depth (μm)	D_60min_	D_900min_	Average Depth (μm)	D_60min_	D_900min_
MSB_0ZrB_2_	42.8	85.5	MSB_1.5ZrB_2_	15.6	57.2
MSB_0.5ZrB_2_	25.1	68.5	MSB_2.5ZrB_2_	12.7	40.3

## Data Availability

No data were generated or analyzed in the presented research.

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
