# Peer review of "Plasma Oscillatory Pressure Sintering of Mo-9Si-8B Alloy with ZrB_2_ Addition"

_materials, 2022, doi:10.3390/ma15072387_

Round 1
Reviewer 1 Report
The Authors investigated the Plasma oscillatory pressure sintering of Mo-9Si-8B composites 2 with ZrB2 addition. The paper is interesting and well written. However few comments on the paper to improve the quality
- Please mention the one or two applications of Mo-Si-B alloy doped with ZrB2in the introduction part.
- On what basis authors selected the wt.% of ZrB2 doping in Mo-Si-B alloy. Please clearly mention. (include the reference)
- It is designed as four components: 0wt%, 0.5wt%, 1.5wt% and 2.5wt% respectively. What is the effect of Mo-Si-B alloy doped with 3% of ZrB2 on hardness of the alloy.
- The Mo-Si-B alloy doped with 2.5wt.%ZrB2 at 9 Hz oscillatory frequency and 1300 degree C shows the improved structure and oxidation resistance film. Please shows some EDS analysis results for evidence.
- Some of the typo errors: (i) In page no 2, Line no. 67 and 68, samples with a nominal composition of Mo-9Si-8B (at.%) were prepared from Mo, Si, and B of 99.9%, 99.5% and 99.5% purities, respectively. Here, 99.5% is repeated twice, please check it.
Reviewer 2 Report
The authors investigated the effect of the addition of ZrB2 to the Mo-9Si-8B alloy on the physical parameters, mechanical properties and resistance to oxidation at the temperature of 1300 o C. For this purpose, the oscillatory sintering technology was used, the authors optimized its parameter (frequency).
I have the following comments.
1. The authors use the terms composite (eg in the title, keywords, less frequently in the text) and alloy (in the text of the manuscript) interchangeably. These are separate concepts. Composites are a combination of two or more materials and the materials remain distinct. Alloys are solid-solutions. So that's the term alloy fits. Please fix this.
2. line 41: ' ..hes?' what does it mean?
3. line 77: 'was 3Hz' why 3 Hz and not 9 Hz?
4. The authors use different markings in the manuscript for the same samples, eg MSB ..., Mo-9Si-8B .., Mo-Si-B -.... It seems to me that it should be arranged so that the markings are unambiguous.
5. line 81: 'The samples were determined by Archimedes method,' It is incomprehensible, what was measured by the Archimedes method?
6. line 82: 'The samples were cut into Φ10 mm×5 mm.' Into what?
7. line 84: 'Kg' > kg
8. line 98: 'in Fig. 4' > in Fig. 3
9. line 106: 'about' > of
10. line 108: 'the Archimedes section method' more details are needed here
11. line 118: 'Figure 4. Optical micrograph about' This is not optical micrograph!
12. Fig. 6., axis descriptions are not readable
13. High temperature exposure details are missing.
14. lines: 205-206: 'The thickness of the oxide layer of the alloy was determined to be 60min and 900min respectively.' min are not a unit of thickness!
15. line 229: 'the oxygen whey" what is this?
16. line 309: year and page information is missing
17. line: 332: page information is missing.
In general, the English language should be carefully corrected.
The scientific approach to the issue under investigation seems appropriate, but the manuscript lacks experimental details that would allow for the repetition of this research.
Reviewer 3 Report
During the review process of “plasma oscillatory pressure sintering of Mo-9Si-8B composites 2 with ZrB2 addition” a few of the queries arose, authors need to reply/rebuttal in revised manuscript.
- Title and abstract is written “ZrB2” instead of ZrB, please update.
- In abstract section please check “oscillatory sintering technology??, Please check it with oscillatory pressure sintering.
- In introduction “Molybdenum is a typical refractory metal with a high melting point of about 2870K, but its oxidation resistance 20 is poor, so it is difficult to be used as a high temperature structural material alone[4].” Please check it the melting point value.
- Figure 3 scale bar must be visual. Please do the needful.
- Figure 4 & 5 horizontal axis need to tech their text format.
- Rewrite the sentence “The theoretical density calculation formula here still adopts the calculation method used in previous research[21]”
- Figure 6 should be improved, in its current form is not clear visual.
- 4 Oxidation behavior of Mo-Si-B alloy at 1300℃ should be modified.
- Please put the citation “This was because in our previous research, high oscillation frequency alloys showed 175 fine structure”
- Figure 8 need to check of Mass loss unit.
- Proofreading should be done carefully.
Round 2
Reviewer 2 Report
The authors corrected the shortcomings I indicated in the previous version of the manuscript. I think the current version is acceptable, although I think a final English proofreading and editing by a professional editor would be beneficial.
Reviewer 3 Report
Author has updated the manuscript in revised form. My decision is Accepted.